# Study of the Landscape Pattern of Shuiyu Village in Beijing, China: A Comprehensive Analysis of Adaptation to Local Microclimate

**Ling Qi [1], Ranqian Liu [2,*], Yuechen Cui [3], Mo Zhou [4], Wojciech Bonenberg [4] and Zhisheng Song [5]**

[1] School of Architecture and Urban Planning, Beijing University of Technology, Beijing 100124, China; 63651106@bjut.edu.cn

[2] School of Architecture, Tianjin University, Tianjin 300072, China

[3] Department of Urban Planning and Design, Faculty of Architecture, The University of Hong Kong, Hong Kong SAR 999077, China; cuicui22332@gmail.com

[4] Faculty of Architecture, Poznan University of Technology, 60-965 Poznan, Poland; mo.zhou@put.poznan.pl (M.Z.); Wojciech.Bonenberg@put.Poznan.pl (W.B.)

[5] Tsinghua University Architectural Design and Research Institute Co., Ltd., Beijing 100084, China; songzhisheng@thad.com.cn

**\*** Correspondence: dories_1012@tju.edu.cn; Tel.: +86-159-1065-0629

**Abstract:** The paper used technical parameters to investigate optimized solutions to protect the ecological environment and improve the microclimate adaptability among the traditional villages in Beijing. Shuiyu Village was used as a case study to analyze the coupling relationship between landscape patterns and the microclimate of traditional villages, with a focus on the ecological relationship between residents and the microclimate. This study also developed a climate index system, which includes computer numerical simulation and microclimate comprehensive analysis methods. The distinct types of landscape patterns were studied using the system. In addition, this paper studied the adaptive design mechanism in-depth, the form parameters of comfort evaluation controllability, and map expression technology of morphological parameters. The findings of this study include the optimized value of the environment based on landscape pattern and the map through the Rhino modeling platform. An interactive platform was developed, and a parametric-assisted optimization design process for traditional villages in the northern part of China was proposed. Moreover, this study concluded optimized strategies and technical guidelines for future planning of the rural areas in northern China with a goal to protect traditional villages and transform them into smart villages with microclimate adaptability.

**Keywords:** adaption; landscape pattern; local microclimate; comprehensive analysis; morphological parameters

## 1. Introduction

The environment of traditional Chinese villages has been severely impacted by globalization, urbanization, and new rural construction, causing its degradation and alienation. One focus of traditional village protection is to ensure the villages' landscape pattern of harmony between humans and nature. The frequently used strategies for quality control of the village environment include using site selection and traditional Fengshui models from ancient times to build landscape patterns. The ancient Chinese generated Fengshui models to make the best use of limited land and established optimal regions, cities and buildings without doing too much damage to the earth [1]. Therefore, their respect for the site, the comprehensive judgment based on "adapting to the situation", and the concept of harmony between human and nature should be critical in the field of sustainable climate adaptability design. Thus, to retain the sustainable development of cities in urban and

rural areas, the protection and development of those traditional villages are necessary. It is critical to study the ecological system in environment construction.

The protection of traditional villages has been a concern in many countries. For example, various research methods and technologies have developed theoretical systems and studied traditional village renewal and spatial patterns in the United Kingdom, Germany, and Japan. Recently, the studies on the landscape pattern of rural settlements have been moved from physical properties to hidden factors in the settlement pattern. In addition, quantitative methods have been used to determine the degree of influence of hidden elements in the spatial distribution [2]. A lot of current research on the development and protection of traditional Chinese villages uses qualitative value research to develop an evaluation system and village spatial form and layout. Moreover, many studies use combined qualitative and quantitative research on building monomers [3,4], and most research studied climate adaptability, human comfort, urban planning and layout, air environment quality, and plant cultivation [3–28]. For example, Liu Binyi and his team used a combination of subjective and objective methods in a case study of the waterfront green space in Shanghai. In their research, green space was studied to improve the comfort feeling in summer. Urban streets were studied as the research object, and the research was conducted from three areas, including the influence of street space on microclimate elements, the evaluation of the comfort feeling, and the interaction between street space and comfort feeling. Comfort evaluation systems with professional characteristics of landscape architecture were established to integrate space elements, microclimate elements, and human feelings [10–13]. Fu Fan et al. undertook research on the indirect effect of urban green spaces on reducing the concentration of fine particulate matter in the air [14] and proposed an optimization plan to improve the thermal environment in Beijing's urban green space system. Dai Fei et al. also studied the impact of the internal spatial structure of green space on the thermal environment and its implementation in Wuhan by analyzing the morphological spatial patterns [15]. Moreover, Feng Xianhui discussed the correlation between plant communities, green space layouts, and microclimate effects in Guangzhou and used the correlation results to optimize site microclimate design strategies and plant community microclimate design [16,17]. Using the microclimate effect, Jin Hexian et al. applied their design strategies among Hangzhou streets and parks [18–20]. Other scholars have been researching the microclimate characteristics of scenic tourist areas, ancient towns, and traditional settlements [23,24]. However, most of the research has been conducted in urban areas [28,29], and limited research has been done on landscape patterns and microclimate adaptation design strategies.

A combination of qualitative and quantitative analysis methods hs been used, including numerical simulation, data comparison, and model construction at the micro-level. Moreover, quantitative, dynamic, and interdisciplinary research has been done in this field, and there are three main problems identified in current traditional village landscape pattern research. Firstly, although many studies are focused on historical and cultural value, settlement spatial distribution characteristics and structure, and architecture, little attention has been paid to the landscape pattern research. Therefore, research in this field is unbalanced, with much focus on macro-scales and micro-research on construction technology but little on the micro-scale research in the environment. Moreover, there is much research on the urban microclimate, but little in traditional villages.

Furthermore, there is limited research on the coupling control of traditional village landscape patterns and microclimate adaptation mechanism characteristics. Second, from the perspective of analysis methods, a lot of qualitative analyses and evaluations have been used, but not many quantitative analysis and design strategies have been used. Thus, there is a lack of comprehensive analysis methods used. Finally, from the perspective of model analysis, design mechanism, and application, most of the existing research conclusions are a single regularity or quantitative result, and the coupling relationship between the multi-factors of microclimate and the characteristics of landscape construction and predictive

control methods have not been investigated further [30,31]. This causes a challenge to transform the evaluation results into practice efficiently and effectively.

To protect the ecological systems and improve the microclimate in the coordinated development of China's Beijing–Tianjin–Hebei coordinated development of urban and rural areas, this paper aimed to use various research methods to investigate internal and external factors in landscape construction and the microclimate of traditional villages that influence the overdevelopment of traditional villages and the destruction of the ideal landscape pattern caused by the misunderstanding of protection. In this study, a case study of Shuiyu Village was conducted in 2017.

## 2. Materials and Methods

### 2.1. Site Description

Shuiyu Village is located in Beigou, Nanjiao Township, Beijing, China (repainted in Figure 1a–c). The village is distributed along a northwest-southeast ditch rock formed in Shuiyugou. The terrain is high in the southwest and low in the northeast, surrounded by mountains. The village was built on a hillside. The landscaped environment of Shuiyu Village is aligned with Fengshui surroundings with mountains and rivers, embracing Yin and Yang (Figure 2). The Zhongjiaoliang in the north is a natural barrier of the landscape. Shuiyu Village is high in the north and low in the south. The Nanpo Ridge is low and gentle, with Shamao Mountain in the east resembling Wu Shamao, and the mountains in the west resembling a throne. The tall mountains in the north block the northwest wind in winter, and with a low south slope, wind can be blown into the valley from the south. The overall ventilation environment is a good condition (Figure 3). The east part of Shuiyu Village preserves the pattern of villages during the Ming and Qing Dynasties, and the texture of streets and lanes is in the Yin-Yang-Bagua pattern. The Kun location is Changling Tuo (appreciating the moon); the trunk location is the big locust tree (Figure 4); the waterfront road is through the village.

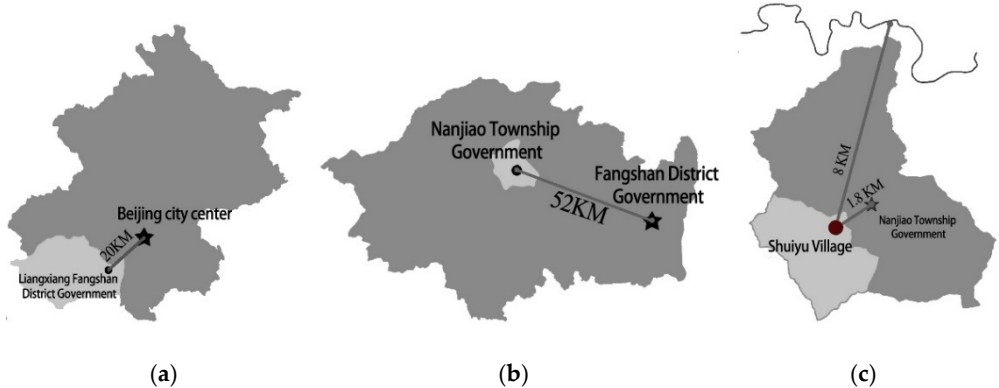

| (**a**) | (**b**) | (**c**) |

**Figure 1.** Shuiyu Village sitemap.

This paper considered the macroscopic combination space of all mountains and water in Shuiyu Village as the research object. It also quantified the coupling relationship between the "shape" of the landscape pattern and the "number" of the microclimate to construct a model [32] and a framework based on microclimate adaptability.

### 2.2. Data Resource

The data source came from three parts: (1) Field observation (measured data of the wind and thermal loop measured data locations of the typical landscape pattern of Shuiyu Village, and location photos with qualitative descriptions); (2) Simulation data (microclimate wind and thermal environment simulation mainly for the terrain of Shuiyu Village); and (3) Parametric terrain data (a digital model presented on the Rhino platform using programming software, such as Grasshopper). This model was developed based

upon the CAD contours of the village and displayed the macroscopic landscape and spatial combination of Shuiyu Village.

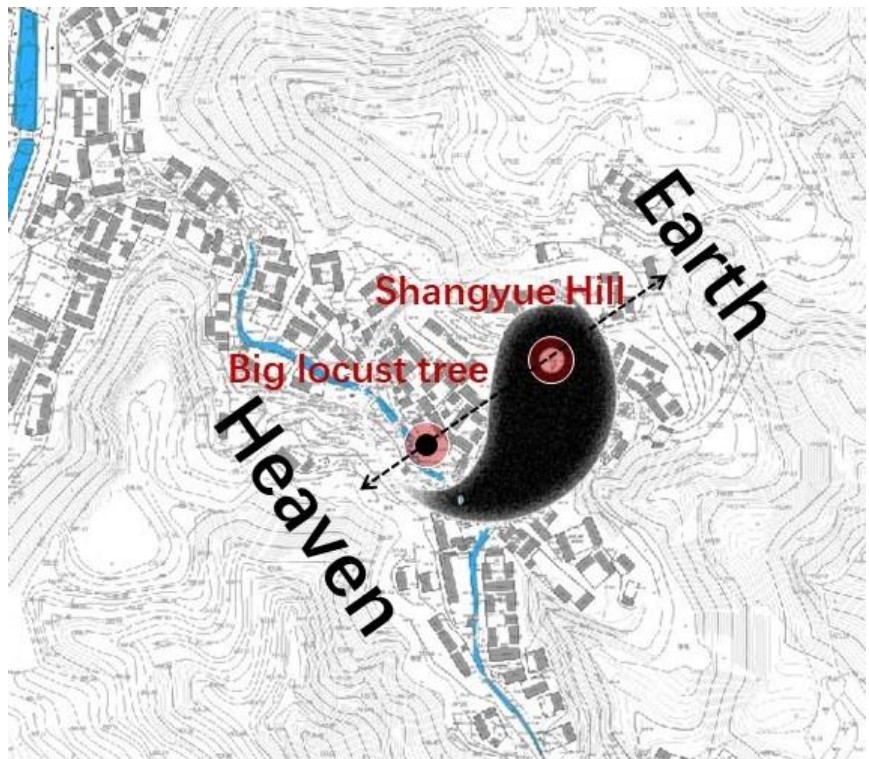

**Figure 2.** The trigram pattern of the east of Shuiyu Village.

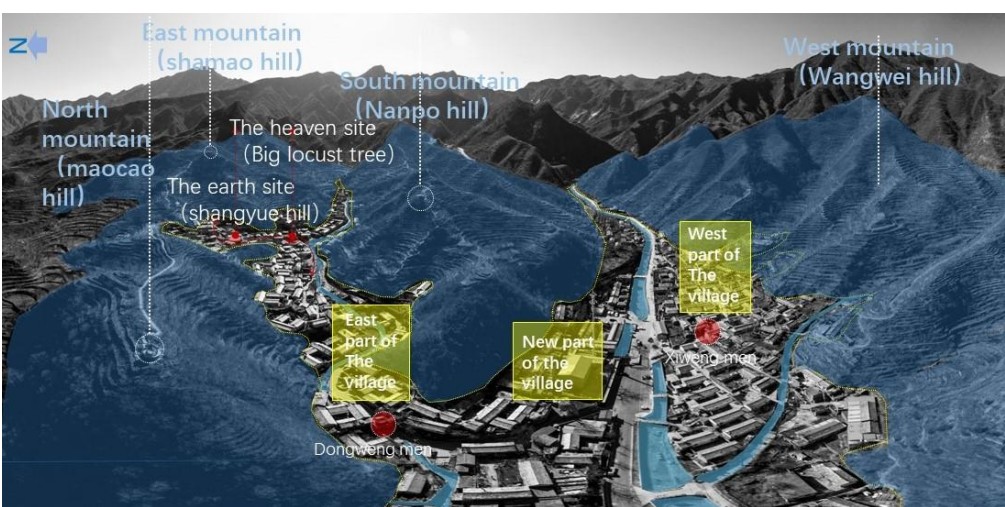

**Figure 3.** Schematic diagram of experimental observation locations.

*2.3. Methodology*

2.3.1. Field Observation

Field observation was conducted 1.5 m from the ground to observe the study object. Climate data, such as air temperature, wind speed and direction, relative humidity, and radiation temperature, at the experimental measurement locations, were analyzed.

(1) Research instruments: The equipment for the experiment included a mobile weather station (ZK-YD6A), hand-held heat-sensitive anemometer (TESTO405-V1, Germany), a temperature and humidity auto-logging instrument (Beijing Tianjianhua Instrument WSZY-1), and a black Bulb temperature recorder (Beijing Tianjianhua Instrument

HQZY-1). The mobile weather station collected data from the fixed space observation locations 24 h a day throughout the year. Due to the limited conditions of the field measurement, one measurement day in each of the 3 seasons and 10 space observation locations were used. The measurement dates were 11 March 2017, 5 July 2017, and 13 January 2018. All the collected data with reference to the data of China Meteorological Network among Beijing area and the data of small weather stations have been used as the input parameters of the developed model. In spring, 18 measuring locations were set up, with nine in each of the east and west villages. Observations were carried out in the mornings and afternoons, respectively. Optimized adjustments were made in summer and winter, and ten measuring locations were set up as well (Figure 4 and Table 1). The measuring locations were selected based on the experimental conditions, landscape environmental characteristics, height and slope orientation, and the principle of uniform distribution of locations. Additionally, the properties of different underlying surfaces, and the overall landscape pattern, buildings, vegetation, and other influencing factors had also been considered.

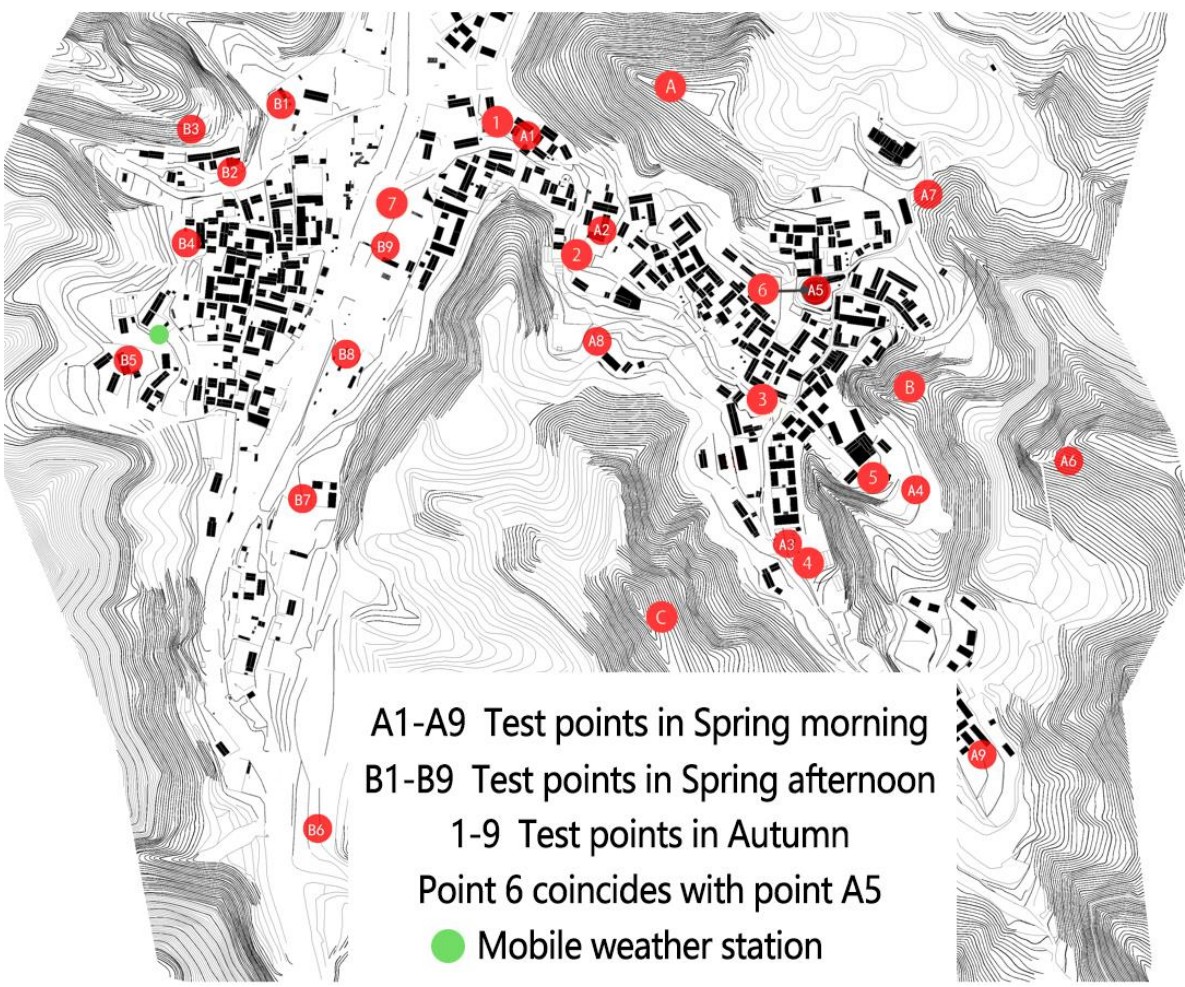

**Figure 4.** The landscape pattern of Shuiyu Village.

(2) Procedure: The research team completed the mobile weather station installation, conducted the 24-h observations at fixed locations, and monitored and collected data through the information platform. The recording frequency of the measured data was 10 min/time. The anemometer was set at the height of about 1.6 m from the ground, and the maximum wind speed was read, and the wind direction was recorded. The temperature and humidity probe was wrapped in tin foil to avoid direct sunlight and placed at a height of about 1.2 m above the ground.

**Table 1.** Experimental Observation Locations.

| Test Locations | Spatial Structure | Environmental Characteristics | Real Scene of the Site Environment | Measuring Instrument | Floor Plan | Landscape Pattern Type |
|---|---|---|---|---|---|---|
| 1 | | The stone road was paved with hills on both sides. The doorway and the valley are in the same direction. | | Hand-held anemometer and temperature and humidity auto logging instrument | | Dependent Slope (east) |
| 2 | | It was located on the hill into the village ramp. The terrain was relatively steep, with mountains and streams on both sides of the southwest and northeast and few dwellings and vegetation. | | Hand-held anemometer and temperature and humidity auto-logging instrument | | Dependent Slope (West) |
| 3 | | The river on the northwest side was frozen. Yang Family Courtyard was on the west side. The underlying surface was slate. | | Hand-held anemometer and temperature and humidity auto-logging instrument | | Open flat ground |
| 4 | | Located at the bottom of the valley. The terrain was narrow, with mountains and buildings on both sides. | | Hand-held anemometer and temperature and humidity auto-logging instrument | | Two sides off the valley (steep) |
| 5 | | Deep in the valley, surrounded by high mountains. Narrow space. | | Hand-held anemometer and temperature and humidity auto-logging instrument | | Three sides off the valley (steep) |
| 6 | | Located on the Moon Viewing Hill, with open terrain, no mountain shelter, no water source, and no vegetation. | | Hand-held anemometer and temperature and humidity auto-logging instrument | | Open flat ground |

**Table 1.** *Cont.*

| Test Locations | Spatial Structure | Environmental Characteristics | Real Scene of the Site Environment | Measuring Instrument | Floor Plan | Landscape Pattern Type |
|---|---|---|---|---|---|---|
| 7 |  | Flat square; adjacent to the main road; two houses with pavilions, surrounded by mountains, no water source, and limited vegetation. |  | Hand-held anemometer and temperature and humidity auto-logging instrument |  |  Open flat ground |
| A |  | Located on the top of the mountain to the northeast of the village. The terrain was high and sunny. Limited vegetation. |  | Hand-held anemometer and temperature and humidity auto-logging instrument |  |  Open mountain top |
| B |  | The pavilion on the top of the mountain that overlooks the ancient village. The wind speed was relatively slow, and the wind direction was relatively stable. |  | Hand-held anemometer and temperature and humidity auto-logging instrument |  |  Open mountain top |
| C |  | The pavilion to the south of Shuiyu Village was open without architectural vegetation. |  | Hand-held anemometer and temperature and humidity auto-logging instrument |  |  Open mountain top |

### 2.3.2. Numerical Simulation

The numerical simulation was undertaken through modeling, using the software of Ecotect and Phoenics. The simulation applied theoretical analysis to model calculations. Numerical simulation involves performing multiple simulation calculations on the same model, checking the measured data, calibrating the measured wind direction, and collecting parameter data, and providing data support for the subsequent parameterized model construction for planning and design.

### 2.3.3. Calculation Method of Microclimate Comfort Index

The layout of traditional village buildings and the landscape environment influences the surrounding microclimate environment and, consequently, the comfort level of the human living environment. The human comfort level in this paper was calculated by using the WBGT balance formula proposed by Dong Liang. The formula calculated the summer microclimate comfort value [33], TS-Givoni index [7], and THI index [34] to retrieve the microclimate comfort value during winter and in spring.

$$\text{WBGT}_{autumn} = 0.8901t + 7.3771 \times 10^{-3}G + 13.8297a - 8.7284v^{-0.0551} \tag{1}$$

$$\text{TS} - \text{Givoni}_{\text{winter}} = 1.7 + 0.1172t + 0.0019G - 0.322v - 0.0073a \tag{2}$$

$$\text{THI}_{\text{spring}} = t - (0.55 - 0.005a)(t - 14.5) \tag{3}$$

$t$: temperature in degrees Celsius, $G$: solar radiation, $a$: relative humidity of the air, $v$: wind speed.

### 2.3.4. Morphological Characterization Quantitative Method

The landscape pattern of traditional villages is affected by natural topographical conditions and human beings' interventional work. It has been continuously reshaped through interactions with the natural environment, forming an ideal landscape pattern. The microscopic morphological representation is reflected in slope, aspect, building orientation, elevation, vegetation coverage, and mountain occlusion. In this study, the morphological characterization of the landscape pattern was measured, and the main parameter factors were extracted by screening the parameterized factors of the geographic spatial model of the landscape pattern. They were presented in the southward space opening and closing degree $X$ and the dominant wind direction opening and closing degree $Y$. The optimization factor extraction uses the sight analysis method to establish the parameterized logic and the space opening and closing degree visualization diagram (Figures 5 and 6). With reference to the related landscape design and parametric views [35,36], this study investigated the slope, aspect, and water body (inundation line) of the digital model of Shuiyu Village. This study also used the traditional village landscape pattern characterization influencing factors and the quantification of related factors to quantify the factors as the linear algebra relationship of the basic operation unit of parameterized programming.

### 2.3.5. Coupling Calculation Method of Landscape Pattern and Microclimate

In this paper, the multivariate linear panel data regression method was used to explore the coupling relationship between landscape pattern and microclimate. In multi-parameter analysis, human comfort had been used as a dependent variable, and landscape morphological characterization factors, such as the opening and closing degree of the southerly space and the opening and closing degree of the dominant wind direction, were taken as independent variables. The contribution of multiple variables to the microclimate was studied, and the regression equation is expressed as follows:

$$W_{it} = \sum_{k=1}^{K} \alpha_{ki}x_{kit} + \sum_{k=1}^{K} \beta_{ki}y_{kit} + \mu_{it} \tag{4}$$

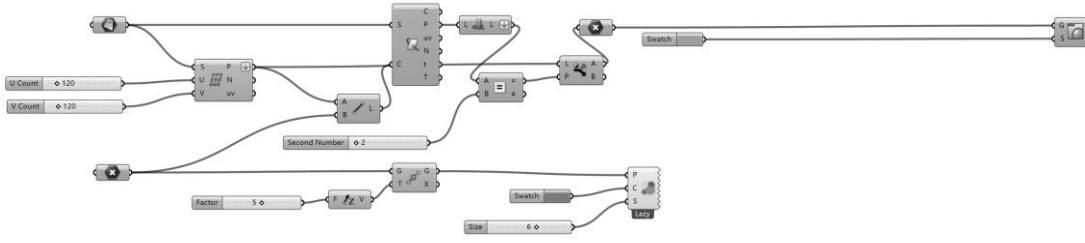

Line-of-sight analysis algorithm

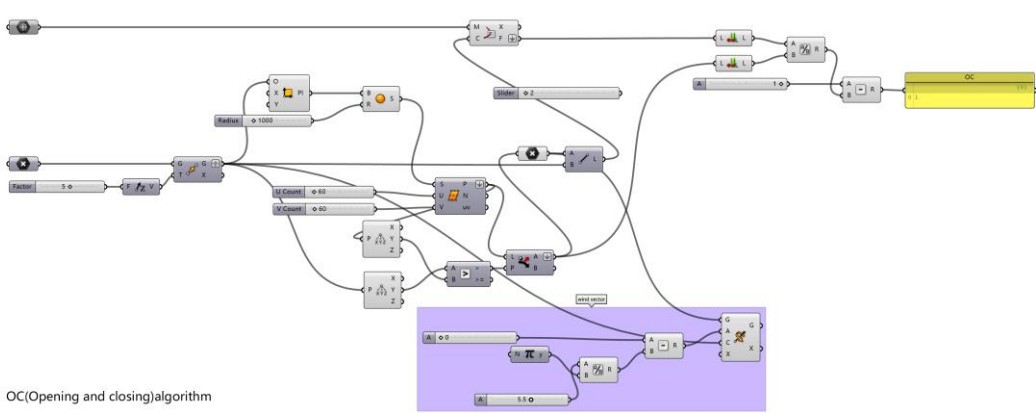

OC(Opening and closing)algorithm

**Figure 5.** Landscape pattern factor algorithm.

Among them, $i = 1, 2, 3, 4, \ldots \ldots, N$, representing locations; $t = 1, 2, 3, \ldots \ldots, T$, representing the time location of the test. $W_{it}$ is the explained variable (human comfort). When the observation value of location $n$ is at $t$, $x_{kit}$ and $y_{kit}$ (south opening and closing degree, dominant wind direction opening and closing) are the $k$-th non-random explanatory variable; locations $\alpha_{ki}$ and $\beta_{ki}$ are the parameters to be measured, and $\mu_{it}$ is a random error.

### 2.3.6. Comfort Evaluation and Visual Expression Method

The microclimate index was rated and evaluated (Figures 7 and 8) using the coupling verification and fitting equation of the landscape pattern factor and microclimate comfort in its microclimate environment and the evaluation standards of human comfort (WBGT index and TS-Givoni index). A visual map of Shuiyu Village's spring, summer, and winter comfort was generated based on climate adaptability.

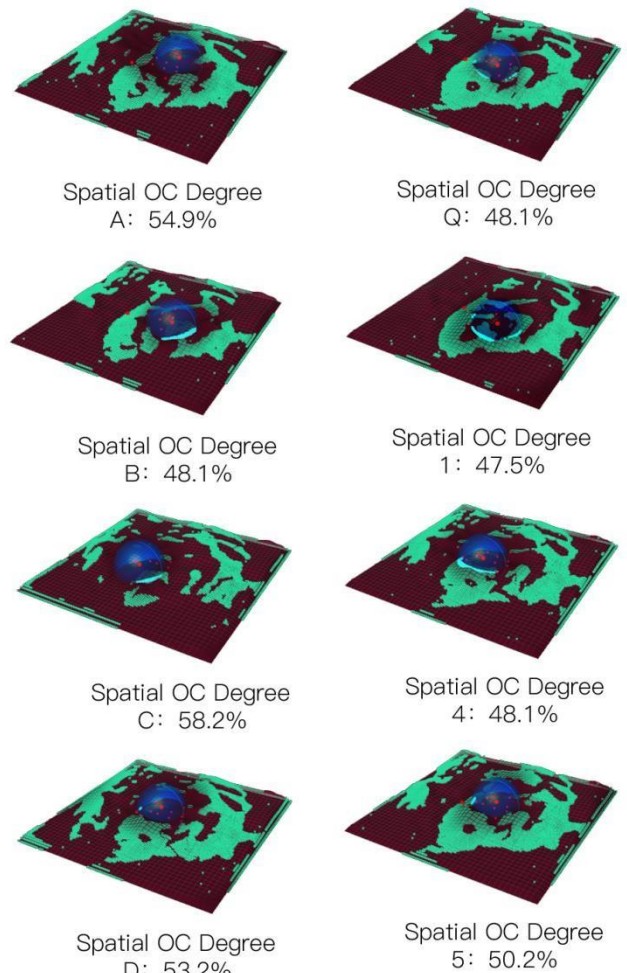

**Figure 6.** Data on the degree of space opening and closing and microclimate factors.

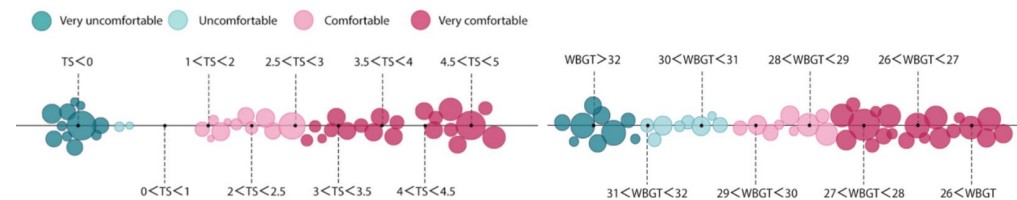

**Figure 7.** Comfort evaluation standard.

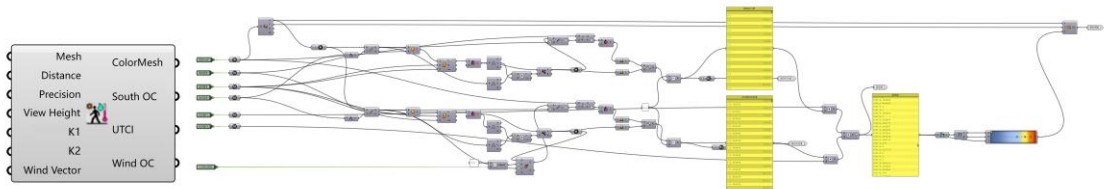

**Figure 8.** Comfort evaluation algorithm.

## 3. Results

### 3.1. Comparison of Microclimate Environment between Shuiyu Village and Beijing

This paper compared the measurement data of Shuiyu Village and the Beijing Meteorological Station. It was found that the average temperature of Shuiyu Village in all four seasons was slightly lower, its humidity was higher, and the wind speed of the village was

lower than that of Beijing City (Table 2). It could be owing to the well-designed landscape and environment of Shuiyu Village.

**Table 2.** Comparison between Shuiyucun Village and Beijing City.

| Time | Average Temperature (°C) | | Average Humidity (RH %) | | Average Wind Speed (m/s) | |
|---|---|---|---|---|---|---|
| | Weather Station | City | Weather Station | City | Weather Station | City |
| 5.11–6.10 | 21.14 | 23.85 | 48.02 | 44.07 | 0.66 | 2.48 |
| 8.11–9.10 | 22.46 | 24.74 | 82.64 | 70.70 | 0.41 | 1.82 |
| 9.11–10.10 | 15.85 | 19.95 | 73.69 | 57.4 | 0.41 | 1.73 |
| 10.11–11.10 | 9.66 | 11.31 | 77.83 | 67.44 | 0.38 | 1.51 |

*3.2. Microclimate Environment in Shuiyu Village*

(1) Temperature: With no wind speed being considered, the higher the openness of the space is, the higher the temperature is. Using the four sets of data in three seasons, the temperature on the top of the mountain and the open area was found higher than the other areas. In spring and summer, the temperature of the location with greater openness raised quickly, and the average temperature was high. During the mornings (10:10 a.m.) in spring, the wind speeds at A6 and A1 were relatively high, causing the rise of temperature at these two locations to slow down. This phenomenon also occurred in summer, but the overall trend remains unchanged. However, in winter, excessive wind speed changed the temperature trend. Consequently, the temperature fluctuations at locations A, B, and C were higher than those of other locations, and Location 3 was located at the high location of the mountain system, and the wind speed was relatively high (Figure 9).

(2) Humidity: Open space affects the sunlight, ventilation, and humidity of the measuring locations and is negatively correlated with the humidity of the measuring locations. It was found that the influence of sunshine was more significant than the influence of wind speed when the spatial opening and closing degrees are similar (Figure 10). The lush degree of vegetation at the measuring location was negatively correlated with the fluctuation of humidity. In particular, the humidity at each measuring location in summer was affected by rainfall. Among the measuring locations, Location 3 was located in the core area of the village, with dense surrounding buildings and plants, which is more conducive to the absorption of rainwater. The humidity rose slowly accordingly.

(3) Wind speed: Wind direction changes when the wind passes through valleys or streets. The wind direction of the experimental locations in the village is parallel to its spatial direction. The wind direction at the top of the mountain in open space is more diverse than the other parts of the mountain. In terms of wind speed, taking Locations 2 and 3 as examples, the altitude and underlying surface of the two locations were similar. However, due to the fact that Location 3 was located at the intersection of north–south and east–west valleys and there was no high mountain around it, but Location 2 was on the south side and sheltered by high mountains, and the openness of the space was low, the average wind speed of Location 2 in the third season was much lower than Location 3. The prevailing wind direction on the measuring day of spring in Shuiyu Village was from the north, and the wind speeds were highest at Locations B1, B2, and B6 on the north windward side, where the building density was small.

The wind speeds at the B83 measuring location were relatively low. Taking the winter measurement results as an example, the wind direction was stable, and the terrain affected the wind speed. The wind speeds were high at the open locations in the north–south direction. For example, the wind speeds at six measuring locations, such as 1, 3, 7, A, B, and C, were relatively higher than the other locations. These locations had at least one opening in the north–south direction, while the wind speeds at other locations were low (Figure 11). The wind speed at the measuring location on the top of the mountain was higher than the wind speed at Location 6 inside the Village. In addition, Location B was the most elevated location in summer and winter, and the wind speed was higher than Locations A and C.

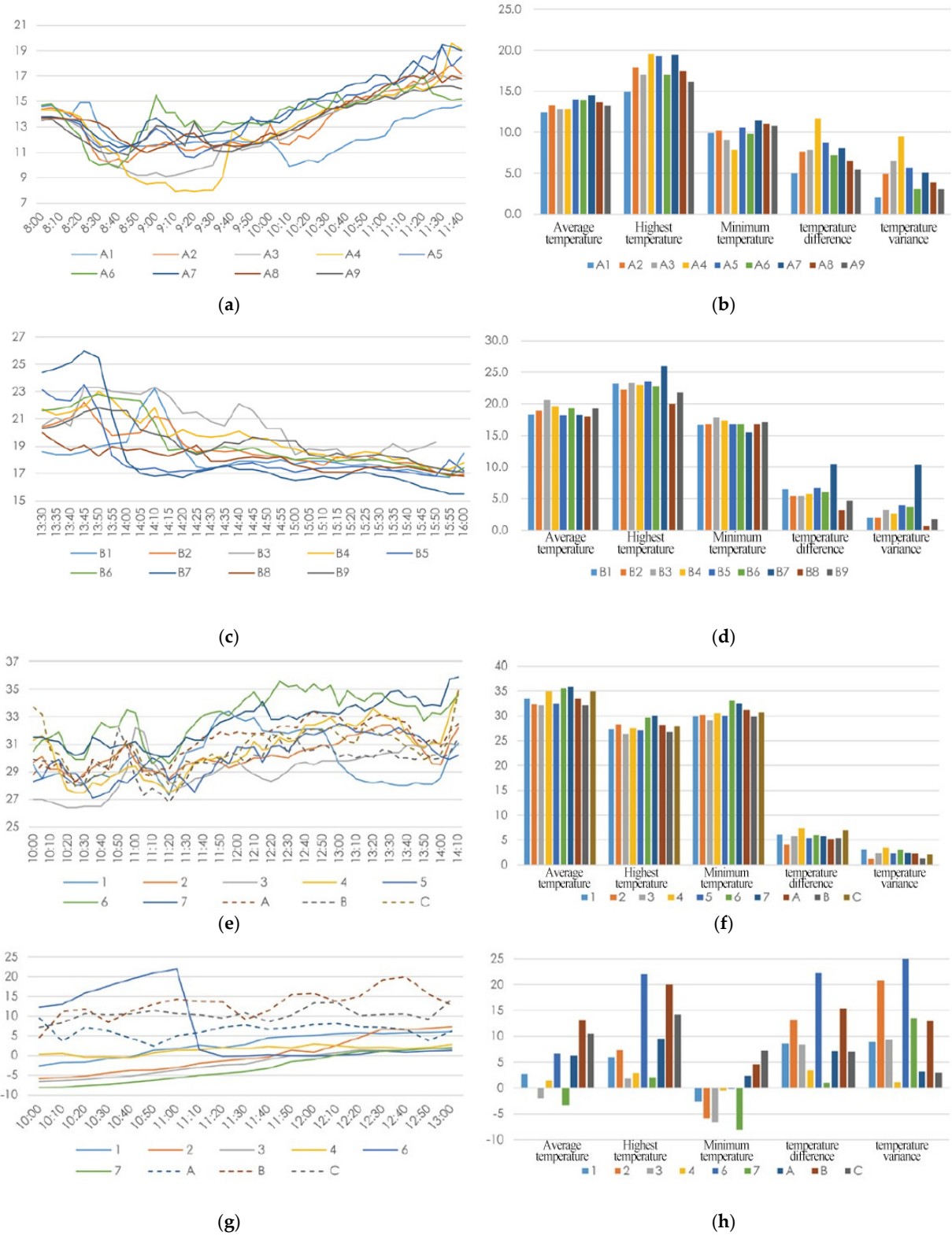

**Figure 9.** (**a**): spring morning temperature curve of Shuiyu village; (**b**): spring morning temperature analysis diagram of Shuiyu village; (**c**): spring afternoon temperature curve of Shuiyu village; (**d**): spring mafternoon temperature analysis diagram of Shuiyu village; (**e**): Autumn temperature curve of Shuiyu village; (**f**): Autumn temperature analysis diagram of Shuiyu village; (**g**): Winter temperature curve of Shuiyu village; (**h**): Winter temperature analysis diagram of Shuiyu village.

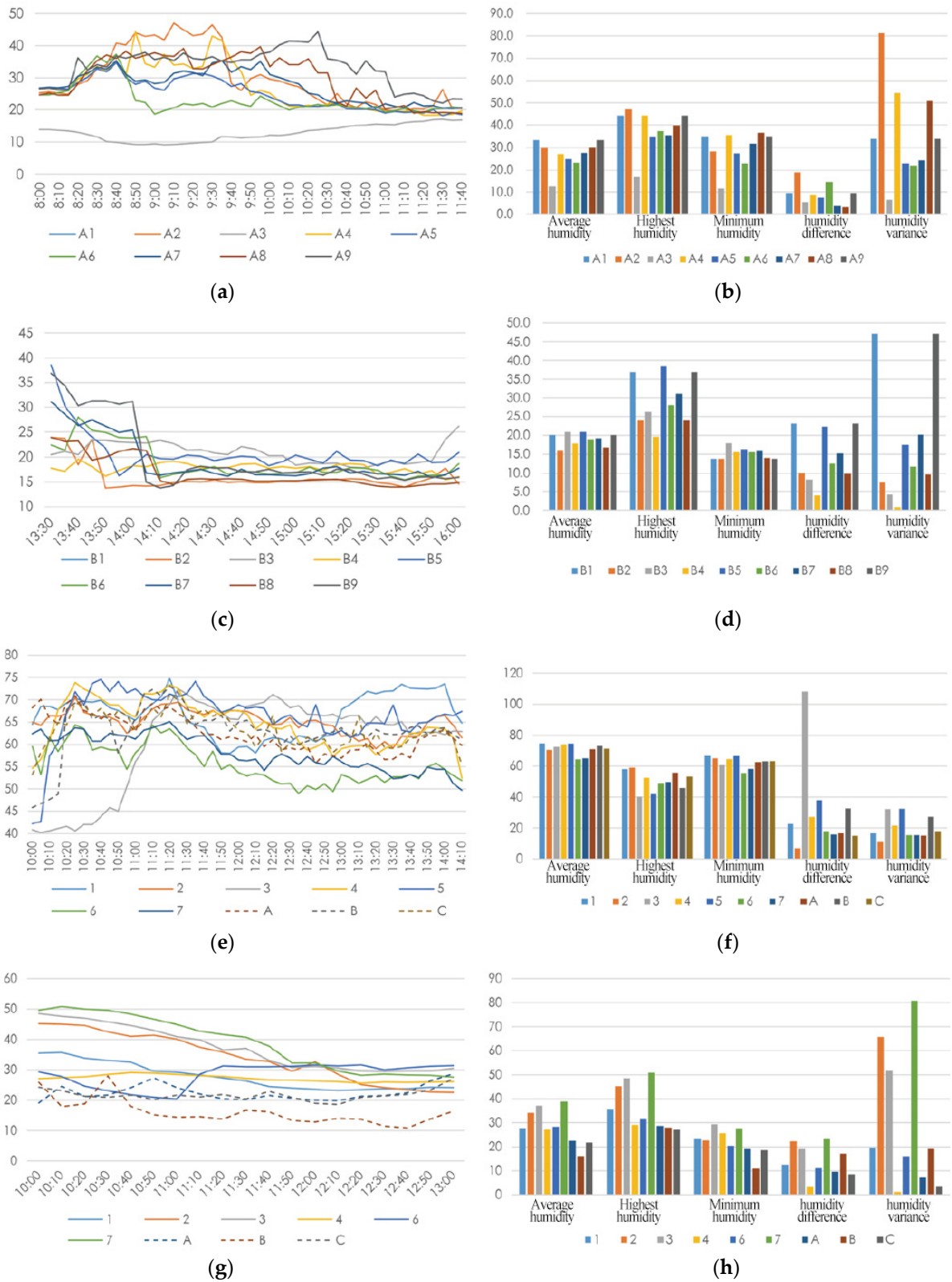

**Figure 10.** (**a**): spring morning humidity curve of Shuiyu village; (**b**): spring afternoon humidity curve of Shuiyu village; (**c**): spring mafternoon humidity analysis diagram of Shuiyu village; (**d**): spring afternoon humidity analysis diagram of Shuiyu village; (**e**): Autumn humidity curve of Shuiyu village; (**f**): Autumn humidity analysis diagram of Shuiyu village; (**g**): Winter humidity curve of Shuiyu village; (**h**): Winter humidity analysis diagram of Shuiyu village.

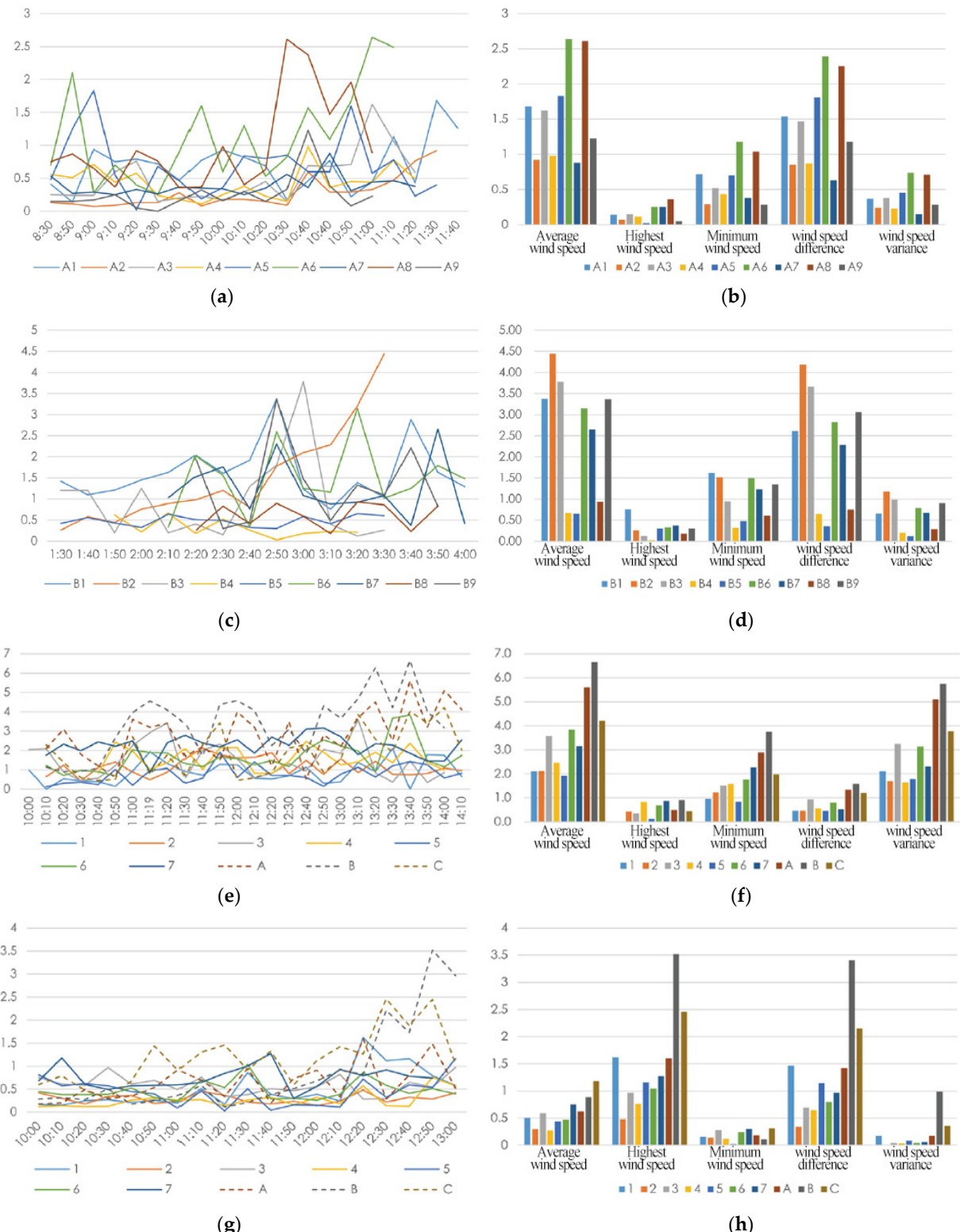

**Figure 11.** (**a**) spring morning wind speed curve of Shuiyu village; (**b**) spring morning wind speed analysis diagram of Shuiyu village; (**c**) spring afternoon wind speed curve of Shuiyu village; (**d**) spring mafternoon wind speed analysis diagram of Shuiyu village; (**e**) Autumn wind speed curve of Shuiyu village; (**f**) Autumn wind speed analysis diagram of Shuiyu village; (**g**) Winter wind speed curve of Shuiyu village; (**h**) Winter wind speed analysis diagram of Shuiyu village.

*3.3. Numerical Simulation Results*

3.3.1. Wind Environment Simulation

The overall wind environment of Shuiyu Village was characterized by high wind speed on the windward side and low wind speed on the leeward side. The East Village is located on the leeward side of the mountain, which is also in a small valley between the two mountains. The Nanshan and Beishan mountain ranges with large slopes pass through both sides of the village entrance. The peaks on the northwest side are barriers to the invasion of the northwest wind, and the south wind passes through the valley mouths from the east and west sides of the Nanshan Mountains. Xicun and Xincun are located in the north–south valley between the three mountains. The West Village was built on the mountains in the valley. The southeast opening of the valley is suitable for summer ventilation and is greatly affected by the valley wind. The mountains on the west side slow down the cold wind from the northwest. These regulate a microclimate environment in winter. However, because the valley is in the southeast direction, lower wind speeds appeared in some areas. Nevertheless, the wind speed at all locations in the east part of Shuiyu Village was balanced (between 2.2 and 2.6 m/s). According to the definition by the Beaufort index, it is within the range between a breeze (1.79 m/s) and a gentle breeze (3.58 m/s), the most suitable wind environment for human settlement. When the breeze entered the mountain valley from an open area, the cross-sectional area of the airflow decreased, and the airflow accelerated, thus forming a strong wind (Figures 11 and 12).

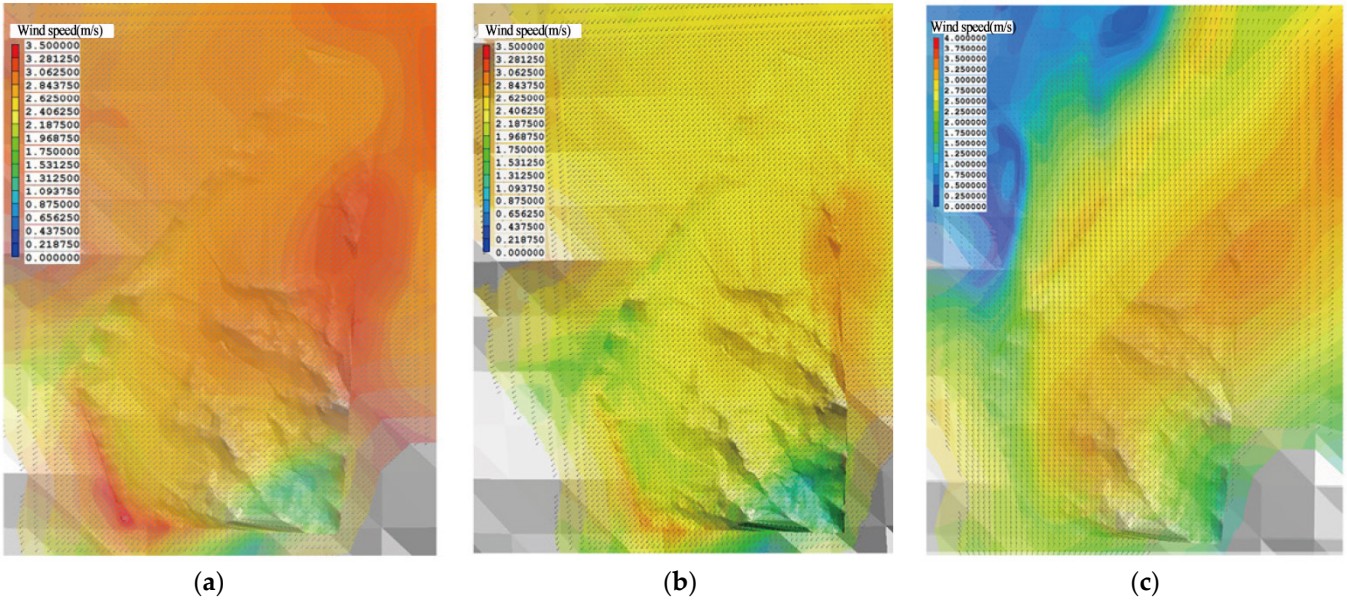

(**a**) (**b**) (**c**)

**Figure 12.** Simulation chart of the Spring (**a**), Summer (**b**), Winter (**c**) wind environment in Shuiyu Village.

3.3.2. Sunshine Environment Simulation

Locations A5 and A7 had the highest temperature because they were exposed to direct sunlight. In addition, the deeper the valley and the smaller the mouth are, the later the sunrise and the shorter the daily sunshine time are. For example, Location A4 was at the deepest part of the valley, the mountain was severely blocked, there was no sunshine before 9:30 am, and the temperature dropped significantly in the afternoon. The location of the mountain had a more significant impact on the sunshine of the area, while the sunlight had a critical influence on the temperature in the site (Figure 13).

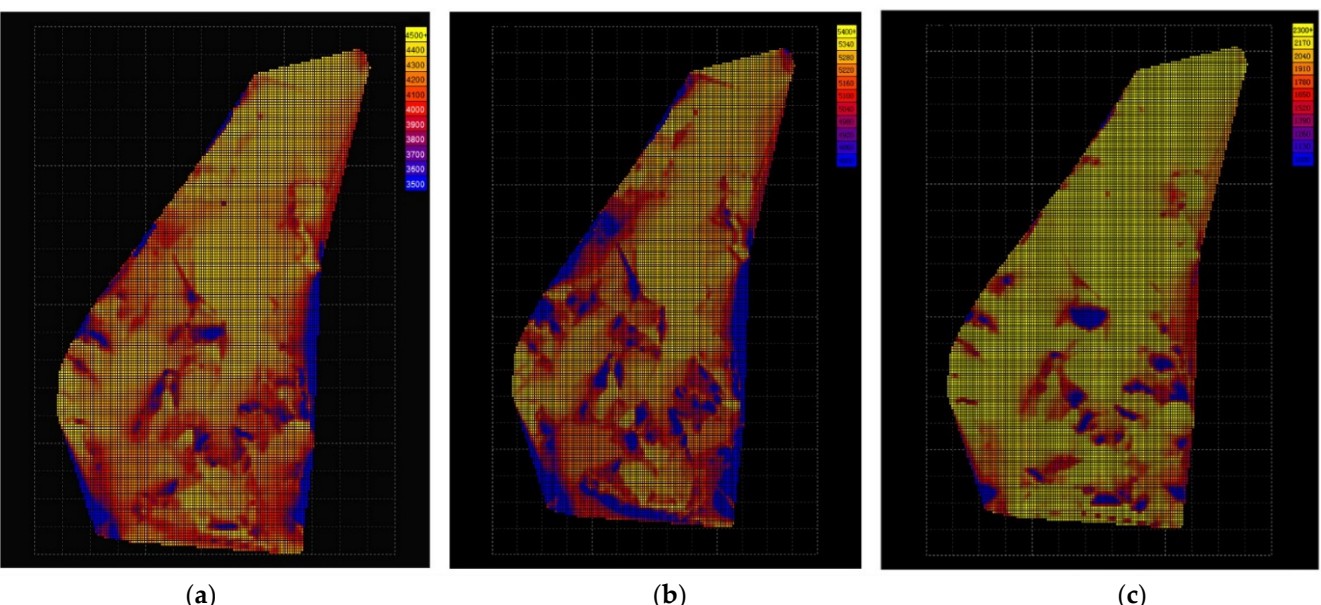

**Figure 13.** Thermal simulation chart of Spring (**a**), Summer (**b**), Winter (**c**) sunshine in Shuiyu Village.

The West Village and New Village are located north–south, and the mountains are on the east and west of the settlement, respectively. The closer they are to the bottom of the valley, the shorter the sunshine time they have. The mountains on the west side of the new village were relatively high, and it affected the sunshine environment of the west village in the afternoon.

### 3.4. Results on the Correlation between Landscape Pattern and Microclimate

Based on the spatial opening and closing degree and the influence degree of the mountain's south side, data from the field measurement location data were collected as the south opening degree and the dominant wind direction opening degree, using multiple linear regression of panel data. The *p*-value determines the significance of the model; that is, whether the landscape pattern has a significant impact on the microclimate environment, and the influence of each index was determined by the estimated parameters (Table 3).

The degree of space opening and closing had a greater impact on summer temperature, humidity, wind speed, and spring temperature. In particular, the microclimate factors in summer were significantly affected by the opening and closing of the space. Among them, the summer temperature was positively correlated with the influence degree of the south side of the mountains. The less sheltered by the mountains on the south side, the higher the temperature. It was negatively correlated with the opening and closing of the dominant wind direction. It was also found that the south side of the mountain had a strong influence on the wind speed in summer. The humidity in summer was positively correlated with the spatial opening and closing degree of the dominant wind direction and negatively correlated with the influence degree of influence of the south side of the mountains. The temperature in spring was not significantly affected by the southward mountain occlusion and was positively correlated with the spatial opening and closing of the dominant wind direction. It was hence affected by the dominant wind direction. Using data collation, the analysis of the landscape pattern was verified.

### 3.5. Expression of Comfort Degree in Shuiyu Village Based on Microclimate Adaptability

The modeling platform was developed by analyzing the relationship between the microclimate and the landscape pattern parameters. Moreover, the comfort grading standard was also included. The village landscape pattern information was used to collect from the locations in the village to develop the human body comfort map. The comfort degree of Shuiyu Village was found to be great in spring, summer, and winter. In the degree map

(shown in Figure 14), values were used to distinguish between colors in spring. Moreover, the comfort evaluation standards were used for selection in winter and summer. The map provided guidance to the adaptive design for the microclimate of a local area.

**Table 3.** Regression models between microclimate and the comfort index.

| Dependent Variable | | Independent Variable | Regression Equation | $R^2$ | *p*-Value |
|---|---|---|---|---|---|
| the comfort index (W) | WBGT (summer) TS-Givoni (winter) THI (spring) | Southward openness = *x* dominant wind direction openness = *y* Southward openness = *x* dominant wind direction openness = *y* Southward openness = *x* dominant wind direction openness = *y* | W = 7.1516X + 33.4031<br>W = −1.0072Y + 36.2758<br>W = −6.7148X + 4.6134<br>W = 0.5019Y + 2.1881<br>W = −0.3009X + 16.1620<br>W = 2.5765Y + 15.1103 | 0.18<br>0.01<br>0.15<br>0.12<br>0.01<br>0.08 | <0.001<br>0.718<br>0.142<br>0.825<br>0.706<br>0.017<br><0.1 |
| Temperature (t) | summer winters pring | Southward openness = *x* dominant wind direction openness = *y* | t = 15.5769X − 9.3404Y + 28.2636<br>t = −88.2864X + 27.1348Y + 24.86<br>t = −0.8629X + 4.6002Y + 15.8701 | 0.25<br>0.13<br>0.08 | <0.05<br>0.1834<br><0.1 |
| Humidity (a) | summer winter spring | Southward openness = *x* dominant wind direction openness = *y* | a = −46.5144X + 22.0111Y + 73.1636<br>a = 54.0313X + 12.8910Y + 5.9255<br>a = 3.1929X + 0.0702Y + 22.1475 | 0.27<br>0.12<br>0.15 | <0.05<br>0.4820<br>0.3583 |
| Wind speed (v) | summer winter spring | Southward openness = *x* dominant wind direction openness = *y* | v = 6.6988X + 0.0753Y − 0.8955<br>v = −0.7197X − 0.3989Y + 1.1853<br>v = 0.4590X + 2.5161Y −0.3912 | 0.37<br>0.08<br>0.05 | <0.05<br>0.6910<br>0.49 |
| Combined | | WBGT (summer)<br>TS-Givoni (winter)<br>THI (spring) | W = 9.5550X − 4.0073Y + 33.9075<br>W = −10.0626X + 3.0953Y + 4.7461<br>W = −0.4472X + 2.6499Y + 15.2445 | 0.22<br>0.13<br>0.09 | <0.001<br>0.1998<br>0.0547<br><0.1 |

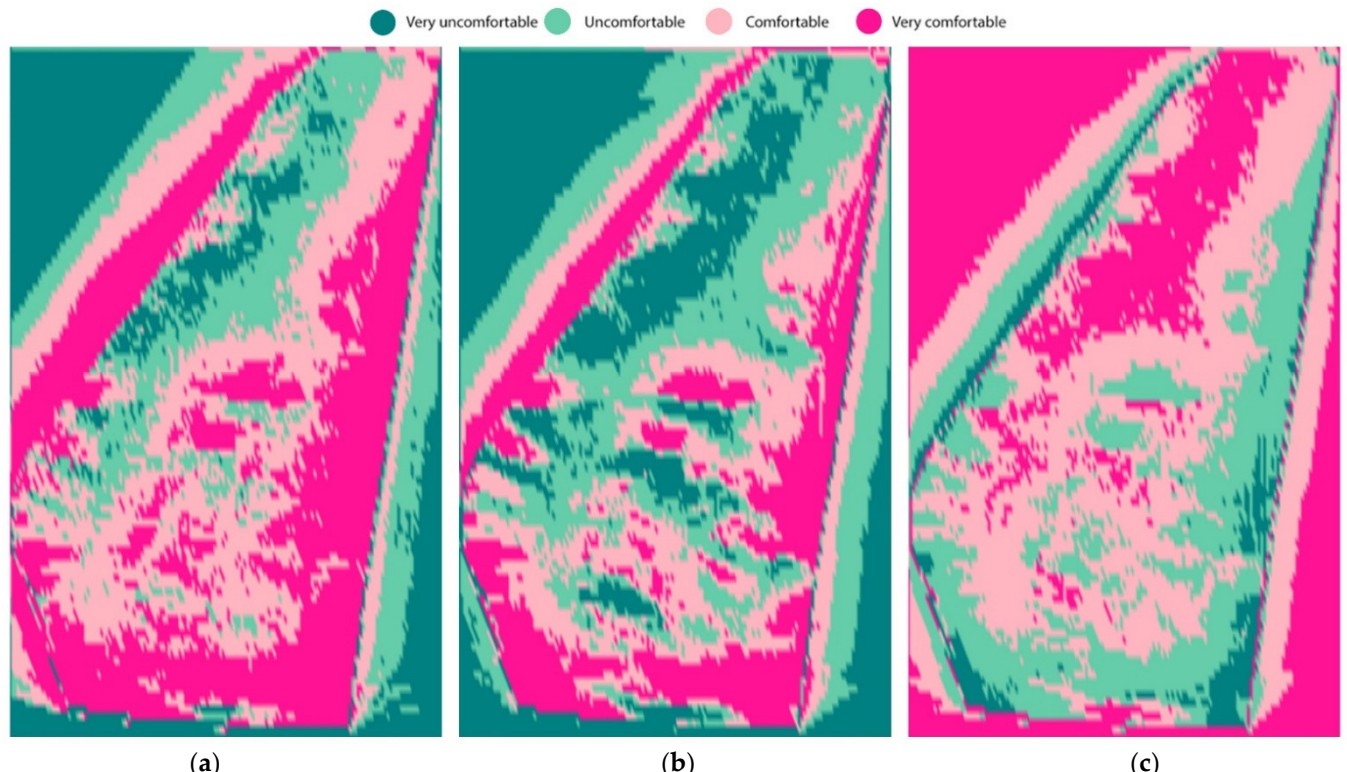

**Figure 14.** Shuiyu Village comfort map in Autumn (**a**), Winter (**b**), Spring (**c**).

## 4. Discussion

### 4.1. Analysis and Discussion

Four conclusions were drawn by using the actual observation measurements, numerical simulation, and discussion of the correlation between microclimate and landscape environment.

(1) It was found that the landscape pattern had a strong relationship with the wind environment simulation results at various locations. The wind was easy to form in the narrow valley area. The wind speed was slow on the leeward of the mountain, but vortex wind might form (e.g., at Location B7).

(2) By simulating and analyzing the heat gain from radiation, it was found that the orientation of the slope of the mountain has a great influence on the heat radiation of the site.

(3) This study found that the environment of Shuiyu Village was suitable for living in spring, summer, and winter, using the data of the sunshine and numerical wind simulation. The simulation indicators of the east part of the village (the ancient village area) show more comfort than the west part. The wind speed in the east was relatively low in spring and winter but high in summer. The intensity of sunshine in the east part of the village was found average in spring, while in summer it was found that more areas were shaded.

(4) The comprehensive analysis of the results of multiple linear regression shows that summer and winter landscape patterns had similar effects on the microclimate comfort when the mountain shelter was considered. Moreover, the southward opening and closing degree and the dominant wind direction opening and closing degree were found correlated to each other. With human comfort in mind, when the summer comfort classification standard was taken into consideration, the greater the opening and closing degree of the dominant wind direction was, the greater the required south opening and closing degree were. The results indicated that Shuiyu Village is a valley type in landscape patterns. When this landscape pattern is open to the south, the dominant wind direction is low (three-sided valley type). Only when the dominant wind direction was high and southward was low (two-sided valleys or four-sided valleys) was it not suitable as a residential location, and the human body comfort was low. When the opening and closing degree of the southward mountain was large, the opening and closing degree of the dominant wind direction would, consequently, be large (such as flat open type or slope-dependent landscape pattern). Then the degree of the southward opening and closing and the magnitude of the dominant wind direction wind speed needed to be considered. When the large mountain environment meets the human comfort index, a reasonable layout of vegetation and buildings can be appropriately used to adjust the human comfort and form a pleasant small site. It was also found that most of the types of landscape patterns met the standards of human comfort. This finding is aligned with the spring environment in North China, when the temperature is suitable for human comfort.

In summary, the microclimate characteristics of different landscape pattern areas in traditional villages were initially affected by the climate and environmental characteristics of the large area; that is, the spatial correlation of the climate environment. North China has a typical temperate monsoon climate with hot and dry summers. In the cold winter, when the environmental characteristics of large areas were similar, the influence of the traditional village landscape pattern was focused. When the general landscape pattern and site selection met the comfort index, the influence of the layout and suitable constructions of buildings were paid attention to, which created the residential environment of the village. Although this model was developed based on the field measuring data of Shuiyu Village and collected the quantitative indicators of the landscape pattern, it provided a certain level of guidance in selecting suitable settlements and the construction and layout of human settlements in the village. Its specific coefficient indicators showed it was not applicable to other villages in non-Beijing–Tianjin–Hebei regions. However, through the verification process of the validity of the model results, the approximate parameterization relationship in the method can be generalized to a certain extent.

Through the algorithmic coupling of "number" (comprehensive microclimate index) and "shape" (optimization of landscape patterns, such as spatial opening and closing), this study found the landscape pattern characteristics of traditional villages (such as the shape of the terrain, the slope of the mountain, the shelter of plants, and the structure of the building) were closely related to local microclimate related factors (such as temperature, humidity, and solar radiation).

This study has some limitations. This study has measured only a number of field locations. This has a limited application to a broader representation of the village landscape patterns and their characterized microclimates for village planning. Moreover, it was found that the impact of some village layouts on microclimate environmental indicators was complex and non-linear. The model was a simplification of the real natural system under certain experimental conditions. In future research, the data richness of the measured database will be increased, with various types of typical landscape pattern locations and different graphic databases of traditional village landscape patterns in different macroclimate environments. Further, different types of microclimate indicator calculation equations will be used to improve the model and collect the parametric coupling state of landscape pattern and microclimate indicators under ideal conditions.

### 4.2. Preliminary Study on Parametric Aided Design Process Based on Climate Adaptability

The study of the parametric relationship between the landscape pattern of traditional villages and microclimate indicators was conducive to digitizing the experience of the original planners in practice. The wisdom of the ancient ancestors and the experience of the planners, such as "Bearing the Yin and Embracing the Yang" and "backing the mountains and facing the water", has been transformed into scientific guidance. The combination of the obtained model and the modeling platform has a broader application prospect, enabling planners to conduct comprehensive research from two dimensions to three dimensions. This paper investigated the applicability and practicability of the model and discussed the application scenarios of the parametric model.

This paper adopted the comfort map expression algorithm combined with the digital model of the site. It was conducted by using the map expression method, guiding the adaptive design of local microclimates, and combining relevant microclimate design experience to formulate optimization strategies and save planning efficiency. This model can be extended in the actual planning of the village. A three-dimensional site model of the village has been used to obtain data from the human body comfort, temperature, wind speed, and slope of the village site. The purpose was to achieve clear, descriptive, and testable primary conditions for the site. It was also aligned with design requirements. Appropriate parameter conditions were set, using the Rhino modeling platform and Grasshopper to iteratively calculate and establish a combination map of morphological relationships to meet the design requirements. A parametric simulation-aided design site selection process was used (Figure 15). The specific process is as follows:

(1) The construction site is investigated to select representative check locations. Then, the microclimate index information is used to generate a digital model of the landscape pattern.

(2) The factors and generated site analytical models are used to analyze the landscape pattern characteristics of the site from the perspectives of elevation, slope, aspect, water environment, and microclimate environment.

(3) According to the construction needs and related conditions (e.g., slope $\leq 10°$, the southeast is relatively open, the wind environment is between the breeze and gentle, and winter and summer conditions are comfortable), the parameters influencing conditions are adjusted to comprehensively analyze factors in the system performance sites. Multiple cycles of adjustment of the parameters are needed to generate and calculate the recommended site selection. It provides an optimized auxiliary design strategy for future detailed traditional village protection and transformation design.

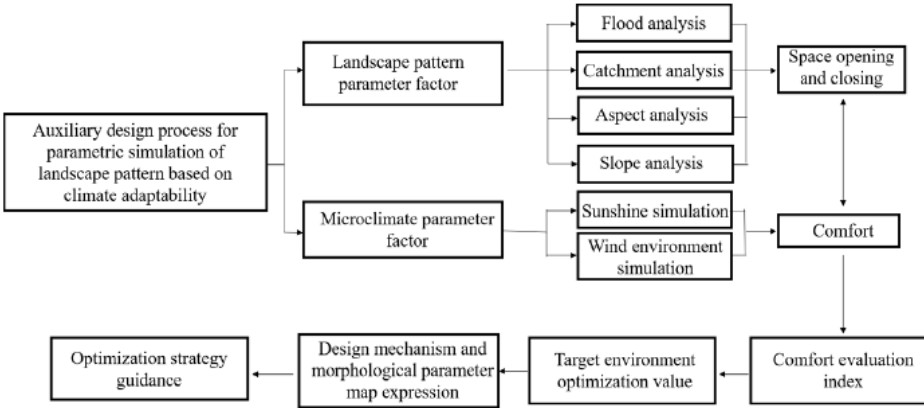

**Figure 15.** Parameterized model construction of traditional village landscape pattern design framework.

### 5. Conclusions

To sum up, this paper developed a parametric model of landscape pattern and discussed the feasibility from the two dimensions of model results and research methods. This model was established based upon the previous case studies of traditional villages and observation measurement and simulated data analysis in Shuiyu Village. Moreover, the algorithmic coupling has been conducted with the parameterized landscape pattern factor morphology generation and the climate adaptability mechanism.

This paper has also made the following contributions. First, it combines the general human comfort evaluation standards and uses Grasshopper to map hierarchical expression. Secondly, it compiles the logic construction process of village comfort evaluation through Grasshopper and R language programming design software. By merging the algorithms, it also tried to build an interactive platform interface schematic and summarize the parametric auxiliary design process in the village planning and site reconstruction. Additionally, it demonstrates the parametric simulation design process from setting parameter conditions to align with the design requirements to evaluate the parametric sites, and finally, generate site recommendations.

The research method used belongs to the "black box" theoretical research; that is, to trace the source from the results. It is a verification study under certain experimental conditions on the ideal landscape pattern of traditional villages. It simulates and quantifies perceptual cognition and inherits traditional villages through quantitative methods. The ecological system provides a scientific underpinning for the improvement of its human settlement environment based upon inheriting and carrying forward traditional culture and greatly improves the efficiency of planning to meet a number of rural planning needs. It also provides a scientific basis for the development of a rural revitalization strategy. The model's applicability has certain limitations due to the limited number of measurement locations. However, the model can be used and applied to any regional characteristics and landscape patterns similar to those of Shuiyu Village.

Although the parametric design strategy was extended based on this model quantifies the "hidden" relationship between the microclimate environment and the landscape pattern through parametric assisted design, it is only limited to the single-line logic of "parameter A→evaluation→screening" [37]. It can be used by designers as an auxiliary technical reference to improve design efficiency. The planning and design of landscape architecture under the human settlement environment science is complicated but systematic. The key is to coordinate and work with the relationship between human beings and nature. Our values determine the position and design process, which consequently determines our design attitude and influences our design method. This ultimately affects the design form and content, designer's values, and ability to make judgments. Our respect for the site and comprehensive judgment based on "adapting to the situation" and "adapting measures to

local conditions" should be critical to our design. Thus, the interactions among the various parameter factors will be investigated in further research.

**Author Contributions:** Conceptualization, L.Q.; Data curation, L.Q., R.L. and Y.C.; Formal analysis, L.Q., R.L. and Y.C.; Funding acquisition, L.Q., M.Z., W.B. and Z.S.; Investigation, L.Q., R.L., Y.C., M.Z. and Z.S.; Methodology, L.Q. and R.L.; Project administration, L.Q., M.Z., W.B. and Z.S.; Resources, L.Q., R.L. and Y.C.; Software, R.L. and Y.C.; Supervision, L.Q., M.Z. and W.B.; Validation, R.L. and Y.C.; Visualization, Y.C.; Writing—original draft, L.Q., R.L.; Writing—review & editing, L.Q. and R.L. All authors have read and agreed to the published version of the manuscript.

**Funding:** This research was sponsored by the International Research Cooperation Talent Introduction and Cultivation Project of Beijing University of Technology(No. 2021C10), the Youth Program of National Natural Science Foundation of China on the project "Based on microclimatological adaptive design of the Beijing-Tianjin-Hebei traditional village landscape pattern research" (No. 51608012), Beijing Municipal Social Science Foundation project (key project) "Building information model technology in the Beijing core area of public facilities design collaborative optimization research" (No. 19YTB050), the National Natural Science Foundation of China on the project "Urban site not permeable form of hydro-ecological effects and its updated design research" (No. 52078006).

**Institutional Review Board Statement:** Not applicable.

**Informed Consent Statement:** Not applicable.

**Data Availability Statement:** Data is contained within the article.

**Conflicts of Interest:** The authors declare no conflict of interest.

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
