# Peer review of "Study of the Landscape Pattern of Shuiyu Village in Beijing, China: A Comprehensive Analysis of Adaptation to Local Microclimate"

_sustainability, doi:10.3390/su14010375_

Round 1

Reviewer 1 Report

Paper sufficiently interesting and clearly structured; a suggestion for the authors could be to never lose sight, in the study of landscape, of the human component. Technological innovation, tools and computer devices allow to quantify and give scientificity to investigations, but the landscape is first of all a social product/construction that can not be relied exclusively on algorithms and mathematical formulas because it would lose its meaning and significance.

Author Response

Thanks for the suggestion, we understand that the landscape is first of all a social product/construction that can not be relied exclusively on algorithms and mathematical formulas , However, in the research of climate adaptability articles, thermal comfort is used as a model to quantify the subjective feelings of the human body, and the human body comfort level of the design scheme is objectively evaluated. so this methods we used in this research is commonly used as technical means. This can not only improve the efficiency of planning, but also generalize qualitative research. Of course, the thermal comfort model also has limitations. For example, subjective human feelings and psychological adjustments will also affect the model. However, due to the limited space of this article and the main research on the ecological wisdom of traditional Chinese villages, this part will not explain in detail.

Reviewer 2 Report

1. Overall, the text has occasional grammatical errors, sentences often have a vague structure, there is absence of dots and commas - so it is not easy to read. So, text needs detailed revision and proofreading  

2. Introduction - the point is clear, but the text has a somewhat vague structure and is not easy to read - revise

3. revise Figure 1-3 - it is not clear where the researched village is.  4. Figure 5 - not clear as well 5. Table 1 - not clear. What is the difference within spatial structure and floor plan? photos are not clear as well. I would not agree to the text under the heading Environmental characteristics really describes the environment characteristics - these are more Spatial characteristics. Or at least describe in methodology what term Environmental characteristics what implies here, maybe even to cite relevant sources. Also, Landscape pattern type is not pointing here to pattern but to the slope of the terrain - is slope of terran = landscape pattern? Since landscape pattern is here object of the paper it should also be explained and referenced to the relevant literature  

Author Response

1. The full text of the article has been revised and proofread in detail
2. The text structure of the introduction has been revised
3. Figure 1-5 has been redrawn, the unreadable parts of the figure have been modified, and the table has been revised again

Reviewer 3 Report

English text corrections are required for many typing errors, for example lines 26, 81,  107, 141, 142...
Line 110, replace the word thesis by paper, or  study
Figures 5 and 6 - texts and numbers are not readable in figures, 
Figures 4 and 5 – it would be beneficial to show the area on the same scale,
Figure 6 it is not clear from which direction is the 3d Picture made
Line 172: selection of 10 points is not readable in Figure  6
In Figures 14, 15, 16,  the location of the village, the measuring points, and also the legend is not readable, so it is difficult to understand how the results are related to the built-up pattern of the village.
For interpretation of results, I recommend using not only the pictures generated from computer simulations – but adding the layers showing and explaining the relation of the landscape pattern and the built-up area of the village.
I recommend expanding the reference section and looking also for studies in international contexts.

Author Response

  1. The full text of the article has been revised and proofread in detail
  2. The text structure of the introduction has been revised
  3. Corrected the errors in the English text and enlarged the text size in Figure 5 and Figure 6.

Reviewer 4 Report

The abstract does not say anything about what is already known about the subject related to the paper. Also, it does not describe the findings of the study. It would be advised to add 2-3 sentences about what is already known and how it is related to the aim of the paper, and to add a section devoted to the results of the analysis applied.

Many figures in the paper are too small – it is not only hard, but impossible to read them. This applies, in particular, to figures 5, 6, 7, 10, 11, 12, and 13. Figure 11 is additionally in Chinese only and therefore difficult to understand for non-Chinese speakers.

It would be recommended to improve English in the paper. It is difficult to understand the intended meaning in some parts of the text, for example in the following sentences: "Then generate a visual map of Shuiyu Village's spring, summer, and winter comfort based on climate adaptability." (245-246), "Parametric model of landscape pattern, and discuss the feasibility of the study from the two dimensions of model results and research methods; based on the model results, extend its application value, the first is to combine the general human comfort evaluation standards and use Grasshopper to map Hierarchical expression, the second is to compile the logic construction process of village comfort evaluation through Grasshopper and R language programming design software." (495-500), and "Demonstrate the parametric simulation design process from setting parameter conditions according to design requirements to parametric site evaluation and then generating site recommendations." (503-505).

Author Response

The findings of the study have been added to the abstract of the paper, and the size of the figures(5,6,7,10,11,12,13) in the paper has been modified to facilitate reading. Correct all Chinese in the picture to English. The English structure and grammar of sentences in the thesis have been revised to facilitate reading.

Round 2

Reviewer 3 Report

The authors significantly improved the article according to the comments of the reviewer.

Author Response

Thanks for suggestion

Reviewer 4 Report

The quality of the paper has been improved. The manuscript can be accepted for publication in Sustainability.

Author Response

Thanks for suggestion